# Protective Effects of Sodium Nitroprusside on Photosynthetic Performance of *Sorghum bicolor* L. under Salt Stress

**DOI:** 10.3390/plants12040832

**Published:** 2023-02-13

**Authors:** Martin A. Stefanov, Georgi D. Rashkov, Ekaterina K. Yotsova, Preslava B. Borisova, Anelia G. Dobrikova, Emilia L. Apostolova

**Affiliations:** Institute of Biophysics and Biomedical Engineering, Bulgarian Academy of Sciences, Acad. G. Bonchev Str., Bl. 21, 1113 Sofia, Bulgaria

**Keywords:** NaCl treatment, nitric oxide, photosynthesis, JIP test, chlorophyll fluorescence, membrane damage, P700 photooxidation

## Abstract

In this study, the impacts of the foliar application of different sodium nitroprusside (SNP, as a donor of nitric oxide) concentrations (0–300 µM) on two sorghum varieties (*Sorghum bicolor* L. Albanus and *Sorghum bicolor* L. Shamal) under salt stress (150 mM NaCl) were investigated. The data revealed that salinity leads to an increase in oxidative stress markers and damage of the membrane integrity, accompanied by a decrease in the chlorophyll content, the open photosystem II (PSII) centers, and the performance indexes (PI _ABS_ and PI _total_), as well as having an influence on the electron flux reducing photosystem I (PSI) end acceptors (REo/RC). Spraying with SNP alleviated the NaCl toxicity on the photosynthetic functions; the protection was concentration-dependent, and greater in Shamal than in Albanus, i.e., variety specific. Furthermore, the experimental results revealed that the degree of SNP protection under salt stress also depends on the endogenous nitric oxide (NO) amount in leaves, the number of active reaction centers per PSII antenna chlorophylls, the enhanced electron flux reducing end acceptors at the acceptor side of PSI, as well as the stimulation of the cyclic electron transport around PSI. The results showed better protection in both varieties of sorghum for SNP concentrations up to 150 µM, which corresponds to about a 50% increase in the endogenous NO leaf content in comparison to the control plants. Our study provides valuable insight into the molecular mechanisms underlying SNP-induced salt tolerance in sorghum varieties and might be a practical approach to correcting salt intolerance.

## 1. Introduction

Climate change is one of the main causes of soil salinization, which in recent years has become a growing problem in agriculture, and leads to a decrease in the area that can be cultivated, and harms plant growth, crop yields, and quality [1,2,3,4,5]. Salinity disturbs the mineral-nutrient relationships, including the uptake and transport of mineral nutrients in plants, which is due to the competition of Na and Cl ions with other ions [6]. It has been suggested that the manipulation of the ion balance (particularly Na^+^/K^+^ homeostasis), combined with selected suitable plant genotypes that improve the ion uptake, is a possible solution to alleviate the effects of salt stress [7,8,9].

High salinity is often coupled with ionic and osmotic stresses in plants, which alters the biochemical and physiological processes in the plants [2,10,11]. Salt stress-induced damage in proteins, lipids and nucleic acids, leads to an increased accumulation of reactive oxygen species (ROS). The excess ROS can cause oxidative damage, and thus a strong inhibition of photosynthetic activity [5]. It has also been shown that salinity modified the contents of several thylakoid membrane polypeptides, including the D1 protein and some polypeptides related to light utilization and oxygen-evolving activity [12,13,14]. The observed alterations in the thylakoid membranes and the photosynthetic functions are connected with a decrease in the density of the photosynthetic structures, the relative size of the plastoquinone (PQ) pool, the electron transport to the photosystem I (PSI) end electron acceptors and the probability of their reduction, as well as to an increase in the thermal energy dissipation [15,16]. Many studies have shown that photosystem II (PSII) is more sensitive to environmental stresses, including salinity, than PSI [17,18,19,20]. In our previous investigations, we have found that salt stress slows down the electron transport between Q_A_ and Q_B_, and that the degree of inhibition depends on the plant species [15,21,22,23]. Moreover, it has also been proposed that the salt-induced damage on the donor side of PSII is greater than that on the acceptor side, and these changes depend on the strength and duration of the stress, as well as the plant genotype [13,24,25].

Nitric oxide (NO) is known as an important signaling molecule in plants under both normal physiological and stress conditions [26,27,28]. NO is a gaseous, diatomic, free-radical with numerous activities, which diffuses readily through the lipid phase of biological membranes [29,30,31]. In higher plants, NO can react with many proteins such as receptors, transcription factors, and signaling molecules [30,32,33]. In chloroplasts, it affects the photophosphorylation, the photosynthetic pigments, the electron-transport chain, the functions of both photosystems, and the oxygen-evolving complex [26,27]. Treatment with exogenous NO, by applying gaseous NO or various NO donors on leaves, leaf discs, intact chloroplasts, or isolated chloroplast membranes, can induce changes in pigment-protein complexes and the photosynthetic parameters of chloroplasts’ membranes under both non-stress and stress conditions [27,34,35]. These NO effects on target sites in the chloroplasts and the photosynthetic apparatus depend on the applied concentration and the time of exposure [27]. 

The critical function of NO in the tolerance to stress induced by oxidative damage is a result of its possibility to scavenge reactive intermediates and halt chain-propagated processes, since it is a free radical [36]. Thus, two ways through which NO may alleviate abiotic stress have been proposed. First, NO may act as an antioxidant, by directly scavenging ROS [37]. Second, NO may act as a signaling molecule in a chain of events that leads to changes in gene expression [30]. Exogenous NO application has been shown to influence plant responses to salt stress [5,28,32,38,39], drought stress [38,40], excess light [41], low and high temperatures [42], ozone, UV-B [42], and heavy-metal stress [37,38,43]. The information on the impact of NO on plants can sometimes be contradictory, because the effects depend on the plant species and seedling age, as well as the duration, and severity of the abiotic stress [31,33,44]. The application of the exogenous NO donor sodium nitroprusside (SNP), has been shown to cause a promising effect on crop germination under control conditions and NaCl-induced stress, as well as on fresh weight, dry weight, and shoot and root elongation [5,45,46,47,48,49]. It has also been shown that NO alleviates oxidative damage (i.e., decreased electrolyte leakage and lipid peroxidation) by increasing antioxidant activities [29,48] and preventing changes in thylakoid shape and size [50]. In various crops, NO has ameliorated the salt stress-reduced leaf area and plant dry matter production [49]. Moreover, under stress, NO boosted the utilization and translocation of many macro- and micro-nutrients, which improved the chlorophyll content [51]. NO-induced alleviation of the toxic effect of high salt concentration results in an enhanced photosynthetic capacity by protecting photosynthetic pigments [48], as well as by quenching extra energy and increasing the quantum yield of PSII [27,34]. A recent study on *Kandelia obovata* plants has demonstrated that the application of SNP (NO donor) under salt stress increased endogenous NO levels, which is connected with reduced ion toxicity, and improved nutrient homeostasis, as well as improved gas exchange parameters and enhanced antioxidant enzyme activities [52].

Sorghum is one of the world’s five major cereal crops [53,54,55], and is an important source of food, feed, and raw material for brewing, so it is expected to be a promising bioenergy crop. It is well known for its high resistance to salt stress and wide adaptability [53]. Therefore, understanding the tolerance responses and strategies in sorghum would be a useful effort to design more salinity-tolerant sorghum genotypes. If the salt tolerance of this cereal crop is increased, it can be widely planted on saline soils, which will be of strategic importance to provide solutions to address energy and food security challenges [9]. 

Considering the protective effect of the NO signaling molecule on the function of the photosynthetic apparatus under salt stress, we hypothesized that spraying with SNP (NO donor) could alleviate the salt-induced damage in crop plants, depending on the applied concentration, and on the plant species and their varieties. For this reason, the aim of this study was to assess the protective effects of different concentrations of SNP (25 μM, 50 μM, 150 μM, and 300 μM) on the functions of the photosynthetic apparatus, the leaf pigment content, and the oxidative stress markers in two varieties of sorghum (*Sorghum bicolor* L. Albanus and *Sorghum bicolor* L. Shamal) under salt stress conditions (150 mM NaCl). The two sorghum varieties (Albanus and Shamal) studied have different sensitivities to salt stress [23,56]. Combined treatment with NaCl and different concentrations of SNP will provide new additional information on the protective effects of NO against salt-induced damage, revealing that NO promotes tolerance to salt stress in a dose-dependent manner. Understanding the ameliorative effect of SNP, as a donor of NO, on the responses of two sorghum varieties towards salt stress may facilitate management strategies for improving their yield in saline soils.

## 2. Results

### 2.1. Pigment Composition

The influence of NaCl (150 mM) alone, and after spraying with different concentrations of SNP (25–300 µM), on the leaf pigment content of the two sorghum varieties studied is shown in Figure 1. The data showed that NaCl treatment resulted in a decrease in photosynthetic pigments (chlorophylls and carotenoids). The total chlorophyll decrease was higher in Albanus than in Shamal (*p* < 0.05), while the decrease of carotenoids was lower in Albanus in comparison to Shamal (*p* < 0.05). After co-treatment with SNP and NaCl, the salt-induced decrease in the chlorophyll content was mitigated but the value remained lower than the control plants in both sorghum varieties studied. These effects were more pronounced at concentrations from 25 μM to 150 μM SNP (Figure 1). The treatment with the highest concentration of SNP (300 μM) caused a smaller prevention of the salt-induced chlorophyll reduction (Figure 1a). Analysis of the carotenoid content showed that all studied concentrations of SNP prevented the salt-induced decrease of these pigments in Albanus. In addition, the better protection in Shamal was registered at 25 μM and 50 μM SNP (Figure 1b).

### 2.2. Oxidative Stress Markers and Injury Index

Determination of H_2_O_2_ and lipid peroxidation (corresponding to MDA content) was used to assess the effects of salt stress (150 mM NaCl) on the two sorghum varieties, as well as the impact of the NO donor SNP on these plants in the presence of NaCl (Figure 2). The amounts of MDA and H_2_O_2_ increased in both Albanus and Shamal after treatment with 150 mM NaCl alone, when compared to the control plants. The data revealed that the increase in both oxidative markers (H_2_O_2_ and MDA) is more pronounced in Albanus than in Shamal (*p* < 0.05). Spraying the plants with SNP (25–150 μM) decreased the content of stress markers (MDA and H_2_O_2_), and in the plants treated with lower concentrations of SNP, these amounts are close to those of the control plants (Figure 2); i.e., low SNP concentrations had a mitigating effect on the NaCl-induced oxidative stress. When treated with the highest concentration of SNP (300 μM) under salinity, the Albanus variety showed an increase in stress markers compared to the control, while in Shamal the values were almost the same as the control plants (Figure 2a,b). 

The histochemical visualization of H_2_O_2_ production in the leaves of the two sorghum varieties indicated that 150 mM NaCl exposure caused severe accumulation of H_2_O_2_ in the whole leaf, while SNP spraying at concentrations of 50 and 150 µM significantly reduced the H_2_O_2_ accumulation (Figure 3).

While MDA was measured as an index of lipid peroxidation, the membrane injury index correlates with the cell membrane permeability, and was used as an additional indicator in identifying the salt tolerance of the plants [57,58,59,60]. The membrane injury index gives information on the damage of the membrane (as a percentage) in comparison to the control plants [61]. The treatment with NaCl alone led to an increase in this parameter, which was more pronounced in Albanus than in Shamal (Figure 4). Moreover, the values of the membrane injury index were higher in Albanus compared to Shamal for all the studied variants (Figure 4). The treatments at all studied SNP concentrations decreased the values of the injury index, with the lowest values being observed for Albanus after co-treatment with 150 µM SNP and NaCl (157% reduction compared to NaCl treated plants), and for Shamal after treatment with 25 µM and 50 µM SNP (202% reduction compared to NaCl treated plants) (Figure 4). An increase in this parameter was observed in both sorghum varieties studied when treated with the highest SNP concentration (300 µM) under salt stress.

### 2.3. Chlorophyll a Fluorescence 

A PAM chlorophyll fluorescence signal analysis demonstrated an influence on the photochemical quenching (qP), the excess excitation energy (EXC), and the non-photochemical quenching (qN) in both examined varieties of sorghum under salt stress (Figure 5 and Figure 6). The values for qP of the control plants were similar in both sorghum varieties. The treatment with 150 mM NaCl led to a greater decrease in the photochemical quenching in Albanus (by 73%) compared to Shamal (by 36%). Foliar treatment with SNP alleviated the harmful effects of NaCl on the photochemical quenching, and the degree of protection depended on the applied concentrations. In addition, the experimental results also revealed that after treatment with the highest SNP concentration (300 µM) and NaCl, the parameter qP had smaller values in comparison to those after treatment with lower concentrations of SNP (25 µM–150 µM) (Figure 5). The data clearly show that SNP spraying with concentrations between 25—150 µM alleviates salt stress-induced changes in qP (Figure 5).

The decrease in photochemical quenching (qP) after treatment with NaCl only corresponds with an increase in the parameters EXC and qN (Figure 6). The values of these parameters were smaller in Shamal than in Albanus for all studied variants except for the treatment with the highest SNP concentration (300 µM) and NaCl. Moreover, the values of EXC and qN were smaller after treatment with 50 and 150 µM SNP and NaCl, in comparison to those after treatment with NaCl alone, for both varieties. The SNP spraying with concentrations between 25–150 µM alleviated salt stress-induced changes in the parameters qP, EXC, and qN, to a greater extent in Shamal than in Albanus (Figure 5 and Figure 6).

More detailed information about the protective mechanism against stress factors, reveals the components of the non-photochemical quenching: energy-dependent quenching (qE), mediated by the proton gradient across the thylakoid membrane; state-transition quenching (qT), caused by reversible phosphorylation of light-harvesting complex (LHCII); and quenching induced by photoinhibition of the PSII reaction center (qI). The effects of different SNP concentrations in the presence of NaCl on these components (qE, qT, and qI) are shown in Figure 7. Under salt stress, significant differences (*p* ˂ 0.05) between the studied varieties were only registered for the photoinhibitory component (qI) (Figure 7a). The component qI strongly increased after NaCl treatment in Albanus, compared to Shamal (no statistically significant differences). Treatment with SNP under salt stress decreased the qI component, but the values of Albanus remained higher compared to Shamal. On the other hand, the effects of SNP on the parameter qT under salt stress were opposite in the two varieties, i.e., qT increased in Albanus and decreased in Shamal. The impact of different SNP concentrations under salt stress on the component qT (state transition quenching) were similar in the respective varieties, i.e., the effects of SNP on qT were independent of the applied SNP concentrations. The data also revealed that the values of qE are similar for all treatments in studied varieties of sorghum, with the exception of the plants treated with 300 µM SNP and NaCl (Figure 7c).

Additional detailed information for the influence of NO on the photosynthetic apparatus under salt stress is provided by the chlorophyll fluorescence induction and selected JIP parameters (ABS/RC, ETo/RC, REo/RC, φPo, Wk, Vj, PI_total_, and PI_ABS_) that were calculated (see Appendix A). Significant differences between the studied parameters for the control plants of the two varieties of sorghum were not registered, except for the parameter PI_total_ (Figure 8). The treatment with 150 mM NaCl influenced the studied parameters, with the effects being stronger for Albanus than Shamal (Figure 8). It should be noted that the greatest differences between the two sorghum varieties after treatment with NaCl were found in the salt-induced changes of the performance index PI _total_, as well as the parameters ETo/RC and REo/RC. Moreover, the data revealed that, when treated with 150 mM NaCl and 25 μM SNP, the values of the studied parameters were similar to those of the control plants (Albanus and Shamal). However, when treated with 50 μM SNP and 150 μM SNP under salt stress, increases in the performance indexes PI_ABS_ and PI_total_, and the numbers of active reaction centers per PSII antenna chlorophyll (RC/ABS) in comparison to the corresponding control plants, were registered, with the effects being greater in the Shamal than in the Albanus variety (Figure 9). 

### 2.4. Redox State of P700

The effects of 150 mM NaCl alone, and co-treatment with SNP (25–300 µM), on the PSI photochemistry were evaluated by the changes in the relative amount of P700^+^ (ΔA/A) and half-time (*t*_1/2_) of dark reduction of P700^+^ (Figure 10). The treatment with NaCl did not influence the parameter ΔA/A for either of the studied variants of sorghum, with changes in the relative amount of P700^+^ (ΔA/A) only being registered after treatment with the highest SNP concentration (300 µM) for Albanus (i.e., a decrease by 38%) (Figure 10a). At the same time, the salt treatment only (150 mM NaCl) led to a strong decrease in the half-time t_1/2_ in both studied sorghum varieties (Figure 10b). The data obtained for Shamal showed that, after co-treatment with NaCl and SNP (for all concentrations), the time *t*_1/2_ was similar to that after treatment with NaCl only. In contrast, for Albanus, the time *t*_1/2_ increased after SNP spraying in comparison to NaCl treatment alone, and the values after application of 25 μM –150 μM SNP were similar to the control plants but decreased after treatment with 300 μM SNP (Figure 10b).

### 2.5. NO Content

The experimental results revealed that treatment with 150 mM NaCl led to an increase in the NO content in the leaves of both studied sorghum varieties. The foliar treatment with SNP led to an additional increase in the NO content in both studied varieties, depending on the applied SNP concentration (Table 1). The data also showed that the amounts of NO in the control and treated plants were greater in Shamal than in Albanus.

## 3. Discussion

Salinity has become one of the main environmental stress factors causing negative effects on plant growth and development, but plants have developed various protective mechanisms to reduce the harmful effects of this stress. It has been shown that an important signaling molecule involved in the protection of plants against abiotic stresses is NO [26,27,29,42,52,62]. This study provides new information on the protective role of SNP, a donor of NO, on the functions of the photosynthetic apparatus under salt stress conditions.

The present study revealed a decrease in the leaf pigment composition after treatment with 150 mM NaCl in both studied sorghum varieties (Figure 1), which could be a result of impaired biosynthesis and/or pigment degradation [63]. In support of this, our hypothesis is the established down-regulation of the genus encoding the proteins of the light-harvesting complexes in sorghum [64]. The data also showed that treatment with 150 mM NaCl caused a more considerable decrease in the chlorophyll content in Albanus than in Shamal. The obtained results are in harmony with earlier investigations on maize, sorghum, tomato, mustard, barley, and pea plants under salt stress [15,65,66,67,68,69,70,71]. Our data (Figure 1) are also in accordance with previous studies on soy bean and pea plants under salt stress [72,73], which have shown that treatment with SNP prevents a salt-induced reduction of the leaf pigment content. SNP application alleviated the salt-induced decrease in chlorophyll synthesis, and enhanced the accumulation of the light-harvesting chlorophyll *a/b* complex of PSII (LHCII) and PSIA/B in barley seedlings [74]. The degree of protection of chlorophylls and carotenoids is different in both studied varieties under NaCl stress, as the SNP spraying protects pigments more effectively at lower concentrations (25 µM–150 µM SNP), while at the highest concentration (300 µM) the protective effect weakens. Similar effects of high SNP concentrations were also observed in barley plants [71]. Moreover, this study showed variations in the effects of SNP on two varieties of wild-type barley, and the authors concluded that 100 µM SNP was most suitable for relieving chlorophylls under salt stress.

It has been found that a high level of salt (Na and Cl) causes the rapid accumulation of excessive ROS, including hydroxyl radicals, singlet oxygen, hydrogen peroxide, etc. [5,75]. Previous investigations have shown that the increased ROS under salt stress causes membrane oxidative damage and enhancement of enzymatic and non-enzymatic defense mechanisms against environmental stresses [23,76,77]. Enhancing the capacity of the antioxidant defense system in plants is the main adaptive response to oxidative stress [78,79]. It was found that the lower oxidative stress in tolerant sorghum seedlings was achieved by significantly increased activities of superoxide dismutase (SOD), ascorbate peroxidase (APX), glutathione reductase (GR), catalase (CAT), and glutathione peroxidase (GPX) compared to the salinity-sensitive cultivar, in response to salt stress [80,81]. Disrupting the balance between normal ROS generation and antioxidant activity leads to oxidative stress in plants. It has been demonstrated that signaling molecules, including NO, stimulate the antioxidant system in plants [78]. The SNP treatment of mangrove species under salt stress was found to increase the endogenous NO levels and reduce ion toxicity by improving the activities of antioxidant enzymes [52]. 

The present study also showed that salt treatment leads to an increase in the H_2_O_2_ content, lipid peroxidation (MDA content), and membrane injury (Figure 2, Figure 3 and Figure 4), which in turn influences the functions of the photosynthetic apparatus. The SNP treatment decreased the levels of H_2_O_2_ (Figure 2 and Figure 3) as a result of modulating antioxidant metabolic pathways and/or acting as a signaling molecule activating ROS-scavenging enzyme activities under salt stress [5,42]. A decreased MDA content (Figure 2) and membrane injury index (Figure 4) after co-treatment with SNP and NaCl, in comparison to NaCl treatment only, indicated that NO can effectively alleviate the salt-induced changes in membranes. Moreover, our data revealed that the degree of protection was different in the two studied varieties, and depended on the applied SNP concentration, with concentrations up to 150 µM SNP giving better protection in both varieties of sorghum, while 300 µM SNP exhibited a weaker protective effect. Similar effects of different SNP concentrations have been shown for barley plants [71].

It has been demonstrated that the changes in the photosynthetic pigments and the membrane damage under salt stress correspond with the disorganized chloroplast grana stacking and instability of the pigment-protein complexes, which leads to an inhibition of the photosynthetic capacity [50,82]. Analysis of chlorophyll *a* fluorescence also provided information on the protective effect of NO for plants under salt stress in the two studied varieties of sorghum. Recently, it has been suggested that the reduction state of the PQ pool (corresponding with qP) is a good indicator to assess the impact of stress factors on PSII’s functionality [83]. The photochemical quenching (qP), indicating the proportion of opened PSII reaction centers, significantly decreased in both varieties of sorghum after treatment with 150 mM NaCl, and in Albanus the effect was stronger than in Shamal (Figure 5). Having in mind our previous study [76], which showed that high salt concentrations restrict the electron flow between Q_A_- and the plastoquinone pool, by influencing Q_A_ reoxidation by PQ and by the recombination of electrons in Q_A_Q_B_- via the Q_A_-Q_B_ ↔ Q_A_Q_B_- charge equilibrium, it could be proposed that there was a greater salt-induced inhibition of these processes in the Albanus variety. The JIP parameters in the present study also revealed salt-induced inhibition of the electron transport flux from Q_A_ to Q_B_ per PSII (ETo/RC), and of the electron flux reducing end acceptors at the acceptor side of PSI (REo/RC), which influenced the performance indexes PI _ABS_ and PI _total_ (Figure 8). All the above changes were also connected with the influence on the number of active reaction centers per PSII antenna chlorophyll (RC/ABS). The data also revealed that the impact on these processes is more pronounced in Albanus than in Shamal. Accordingly, under salt stress, the Albanus variety showed higher PSII excess energy (EXC) (Figure 6b) and non-photochemical quenching of chlorophyll fluorescence (qN) compared to the Shamal variety (Figure 6a). It is well known that non-photochemical quenching of chlorophyll fluorescence is a photoprotective mechanism in plants and involves three components: photoinhibitory quenching (qI), state-transition quenching (qT), and energy-dependent quenching (qE) [84]. Our data revealed some differences between both studied varieties of sorghum under salt stress, mainly in the qI (Figure 7a). Lower values of photoinhibitory quenching after treatment with 150 mM NaCl (with and without SNP) were observed in the Shamal variety compared to Albanus, i.e., changes in qI are variety-specific (Figure 7a). Having in mind that qI is mainly induced by the inactivation of PSII reaction centers [85,86], it could be suggested that the variations in qI between the studied varieties assume the different salt sensitivities and salt damage of the PSII complexes in them. The SNP treatment caused an increase in state-transition quenching (qT) in Albanus and a decrease in Shamal, when compared to NaCl treatment alone (Figure 7b). This parameter depends on the shift of the light-harvesting complex from PSII to PSI (state transition). The component qE was not changed, or was only slightly increased, at 300 µM SNP.

Since chlorophyll a fluorescence is often used as an indicator of changes in the photosynthetic apparatus under different abiotic stress factors [15,87,88,89], the data in the present study suggest an influence of high salt concentration on the functions of the photosynthetic apparatus. The salt-induced changes in studied parameters of the chlorophyll a fluorescence diminished following NO application in both varieties of sorghum. Foliar spraying with SNP under salt stress led to an increase in the open PSII reaction centers (i.e., qP increases) (Figure 5), the improvement of the electron transport from Q_A_ to Q_B_ per PSII (ETo/RC), and the electron flux reducing end acceptors at the acceptor side of PSI (REo/RC), as well as to an increase in the PI_ABS_ and PI_total_ (Figure 8 and Figure 9). At the same time, in both studied varieties of sorghum, SNP decreased the excess excitation energy (EXC), photochemical quenching (qN), and especially the photoinhibitory component qI (Figure 6 and Figure 7a). A similar impact of SNP under salt stress on photosynthetic performance has been shown for citrus seedlings [90], *Solanum melongena* L. [35], and mustard [69]. In addition, our data also revealed that SNP concentrations up to 150 µM provide better protection against salt stress, which corresponds with endogenous leaf NO content of 63–74 nmoles NO g^−1^ FW for Shamal, and 45–57 nmoles NO g^−1^ FW for Albanus (Table 1). These results suggest that the optimal amount of NO for better protection of plants varies between the two varieties. It is revealed that better protection of the function of the photosynthetic apparatus corresponds with an increase in the numbers of active reaction centers per PSII antenna chlorophylls (RC/ABS) (Figure 9), as a result of enhanced chlorophyll synthesis and changes in LHCII [74]. It could be suggested that this is one of the reasons for better photosynthetic performance in the studied varieties of sorghum after foliar application of SNP under salt stress.

It is well known that PSI is more stress-resistant than PSII [19]. Our data revealed that PSI photochemical activities, after treatment with 150 mM NaCl, were similar to the untreated plants for both varieties of sorghum (Figure 10a). At the same time, the salt treatment caused a decrease in half times t_1/2_, indicating an increase in the cyclic electron transport around PSI, which prevents the photosynthetic apparatus from stress-induced oxidative damage [91]. After the co-treatment with SNP and NaCl, the photooxidation of P700 was not affected except at a concentration of 300 µM for the Albanus variety. The data demonstrated that all studied SNP concentrations under salt stress stimulate cyclic electron transport around PSI (i.e., t_1/2_ decreased) in Shamal, which could be a reason for the better protection of the photosynthetic performance in this variety compared to Albanus.

## 4. Materials and Methods

### 4.1. Plant Growth Conditions and Treatments

Two varieties of sorghum, with different sensitivities to salinity *(Sorghum bicolor L.* Albanus and *Sorghum bicolor* L. Shamal), were used in this study. The seeds were kindly provided by Euralis Ltd. (Lescar, France), a large company that develops cereal hybrids for Europe. The seedlings were cultivated after germination in half-strength Hoagland solutions containing: 2.5 mM KNO_3_, 2.5 mM Ca(NO_3_)_2_, 1 mM MgSO_4_, 0.5 mM NH_4_NO_3_, 0.5 mM K_2_HPO_4_, 23 μM H_3_BO_3_, 4.5 μM MnCl_2_, 0.4 μM ZnSO_4_, 0.2 μM CuSO_4_, 0.25 μM Na_2_MoO_4_, and 20 μM Fe-EDTA (pH 6.0) [23]. The seedlings were grown under regulated conditions, including a 12 h light/dark photoperiod, a light intensity of 150 µmol photons m^−2^ s^−1^, 28 °C (daily)/25 °C (night) temperature, and 60% humidity. After 14 days of growth, the two sorghum varieties were sprayed with different concentrations of the NO donor SNP (25 μM, 50 μM, 150 μM and 300 μM) 24 h before the addition of 150 mM NaCl in Hoagland nutrient solution. The combined effects of NaCl and SNP were assessed after 6 days of exposure. The following variants were studied: control (0 μM SNP and 0 mM NaCl); NaCl (0 μM SNP and 150 mM NaCl); 25 SNP + NaCl (25 μM SNP and 150 mM NaCl); 50 SNP + NaCl (50 μM SNP and 150 mM NaCl); 150 SNP + NaCl (150 μM SNP and 150 mM NaCl); 300 SNP + NaCl (300 μM SNP and 150 mM NaCl. Two independent experiments were performed. The measurements and analyses were performed on mature expanded leaves.

### 4.2. Pigment Composition

The pigments were extracted from the leaves using ice-cold 80% (*v*/*v*) acetone. Details of the pigment analysis are given in Stefanov et al. [23]. Total chlorophylls (Chl *a+b*) and carotenoids (Car) were determined spectrophotometrically using a spectrophotometer (Specord 210 PLUS, Edition 2010, Analytik-Jena AG, Jena, Germany) using Lichtenthaler’s equations [92]. 

### 4.3. Oxidative Stress Markers and Membrane Injury Index

The determinations of hydrogen peroxide (H_2_O_2_) and the level of lipid peroxidation, estimated by the amount of malondialdehyde (MDA), were performed as previously described in [23,93]. The amount of H_2_O_2_ was evaluated using trichloroacetic acid, as suggested by Alexieva et al. [94] The absorption was measured at 390 nm (Specord 210 Plus, Edition 2010; Analytik Jena AG, Jena, Germany) and using the molar extinction coefficient 0.28 μM^−1^ cm^−1^. The amounts were expressed as nmol per g dry weight (DW). The MDA level was determined using thiobarbituric acid, as described by Heath and Packer [95]. The quantity of MDA was calculated as nmol per g DW using the absorbance at 532 nm (Specord 210 Plus, Edition 2010; Analytik Jena AG, Germany) and the molar extinction coefficient 0.155 µM^−1^ cm^−1^. For each variant, frozen leaf samples were collected from 10 different plants. 

The leaf cell membrane injury index was estimated as described previously in Kocheva et al. [61]. Mature leaves from different plants were cut into pieces (2 cm in length) and then incubated in tubes with 25 mL distilled water for 24 h at room temperature. Then the electrical conductivity of the solutions was measured using a conductometer (Hydromat LM302, Witten, Germany). After that, the samples were boiled for 30 min and cooled to room temperature to determine the final electrical conductivity. The injury index values were calculated as: I (%) = [1 − (1 − T_1_/T_2_)/(1 − C_1_/C_2_)] × 100, where T_1_ and T_2_ are the first and second (after boiling) conductivity measurements of the solutions with the treated plant leaf samples, while C_1_ and C_2_ are the respective values for the untreated controls [61].

Diaminobenzidine (DAB) staining was performed to visualize H_2_O_2_ accumulation in leaves, as previously described in Daudi and O’Brien [96]. For DAB staining, fresh leaves were soaked in 1 mg mL^−1^ DAB solution and then incubated in the dark at room temperature for one night. Peroxidase catalyzed the reaction of DAB with H_2_O_2_ to form a brown polymer. After that, the leaves were soaked in boiling 95% ethanol to remove the background.

### 4.4. Room Temperature Chlorophyll Fluorescence

For measurements of the pulse-amplitude chlorophyll a fluorescence (PAM), a PAM fluorometer (PAM 101–103, Walz GmbH, Effeltrich, Germany) was used. Before measurements, the leaves were dark adapted for 20 min. The maximal fluorescence levels Fm (for the dark-adapted state) and Fm’ (for the light-adapted state) were measured using a saturated pulse illumination (3000 µmol photons m^−2^ s^−1^, 0.8 s) produced by a Schott lamp KL 1500. (Schott Glaswerke, Mainz, Germany). The intensity for determining the minimal fluorescence level (F_0_) in darkness was 0.04 µmol photons m^−2^ s^−1^. The actinic light intensity was 150 µmol photons m^−2^ s^−1^, similar to the light growth condition. The following parameters were estimated as previously described [97,98,99,100,101]: qP—the photochemical quenching coefficient; EXC—the excess excitation energy; qN—the non-photochemical quenching coefficient; qE—the energy-dependent quenching component; qT—the state-transition quenching component, and qI—the photoinhibitory quenching component of the non-photochemical quenching.

Chlorophyll fluorescence induction curves were measured using a Handy PEA+ instrument (Hansatech, Norfolk, UK), as described by Stefanov et al. [15,23]. The leaf samples were dark-adapted for 20 min at room temperature. The intensity of the light pulse was 3000 µmol photons m^−2^ s^−1^. All examined variants demonstrated an increase in multiphase chlorophyll fluorescence during the first second of illumination after dark adaption. The measured signals were used to calculate some JIP parameters: the absorption flux per RC (ABS/RC), the numbers of active RC per PSII antenna chlorophyll (RC/ABS), the electron transport flux from Q_A_ to Q_B_ per PSII (ETo/RC), the electron flux reducing end acceptors at the acceptor side of PSI (REo/RC), the maximum quantum yield of primary photochemistry (φPo), the relative variable fluorescence at the J step (Vj), the ratio of K phase to J phase (Wk), and the performance indexes PI _ABS_ and PI _total_ [102,103,104,105,106] The details for the JIP parameters are given in Appendix A. 

### 4.5. P700 Photooxidation

The P700 redox states were measured in vivo using a PAM-101/103 fluorometer equipped with a dual-wavelength (830 nm) unit (Walz ED 700DW-E), as in Stefanov et al. [23]. The dark-adapted detached leaves were illuminated with a far-red light, supplied by a photodiode (102-FR, Walz GmbH, Effeltrich, Germany). The far-red light-generated absorbance changes at 830 nm (ΔA/A) were used to examine the changes in the oxidation of P700 (P700^+^) and the half-time of P700^+^ dark reduction (*t*_1/2_) [107]. 

### 4.6. NO Content

The amount of NO was determined as previously described by Fatma et al. [69]. The leaves were homogenized with an acetic acid buffer with low pH, containing zinc acetate. The homogenate was centrifuged. The supernatant was neutralized by adding charcoal and then Greiss reagent was added. The determination of the NO amount (nmol g^−1^ FW) was carried out from a calibration curve plotted using sodium nitrite as a standard.

### 4.7. Statistics

The mean values (±SE) were calculated from two independent treatments with four replicates of each variant, and the statistically significant differences were determined using a one-way ANOVA analysis. Mean values were considered statistically different after Tukey’s significant difference post-hoc test by using the Origin 9.0 software (OriginLab, Northampton, MA, USA). 

## 5. Conclusions

In the present study, more detailed information about the SNP impact under salt stress on the primary photosynthetic processes in sorghum was shown for the first time. The data clearly showed that parameters characterizing the functions of PSI (photooxidation of P700) and PSII (chlorophyll fluorescence analysis) were similar in the studied varieties of sorghum at physiological conditions. The functions of the photosynthetic apparatus were less influenced in Shamal than in Albanus, indicating that the Shamal variety is more salt tolerant than Albanus. Foliar treatment with SNP (a NO donor) alleviated the NaCl toxicity on the photosynthetic apparatus, and its effects are concentration-dependent and variety-specific. Nitric oxide mitigates the adverse effects of salt stress to a greater extent in Shamal in comparison to Albanus, which is accompanied by (i) higher amounts of NO in the leaves (under salt stress and physiological conditions), (ii) a bigger number of active PSII reaction centers per antenna chlorophylls; (iii) the improvement of electron flux reducing end acceptors at the acceptor side of PSI, and (iv) an increased capacity for cyclic electron transport around PSI. In addition, the data showed that SNP concentrations up to 150 µM SNP have better protective effects in both studied varieties of sorghum, which corresponds with an up to 50% increase in the endogenous leaf content of NO in comparison to that in the control plants. The results presented in the current study could provide a promising approach for improving the salt tolerance of sorghum varieties in salinity-affected areas.

## Figures and Tables

**Figure 1 plants-12-00832-f001:**
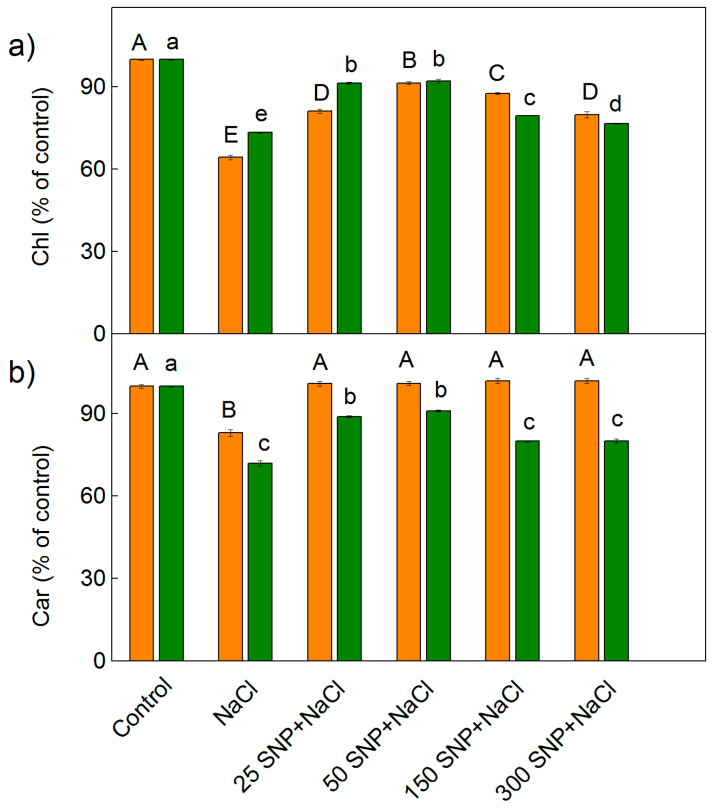
Effect of SNP on (**a**) the leaf total chlorophyll (Chl) and (**b**) carotenoid (Car) content of two sorghum varieties (Albanus—orange bars and Shamal—green bars) under salt stress. The values are expressed as a percentage of the respective control. The control values (100%) for Albanus are: Chl = 25,471 μg g^−1^ DW, Car = 4854 μg g^−1^ DW; for Shamal are: Chl = 24,854 μg g^−1^ DW, Car = 5450 μg g^−1^ DW. Mean values (±SE) were calculated from eight independent measurements. The significant differences among treatments at *p* < 0.05 are indicated by different letters (uppercase for Albanus and lowercase for Shamal).

**Figure 2 plants-12-00832-f002:**
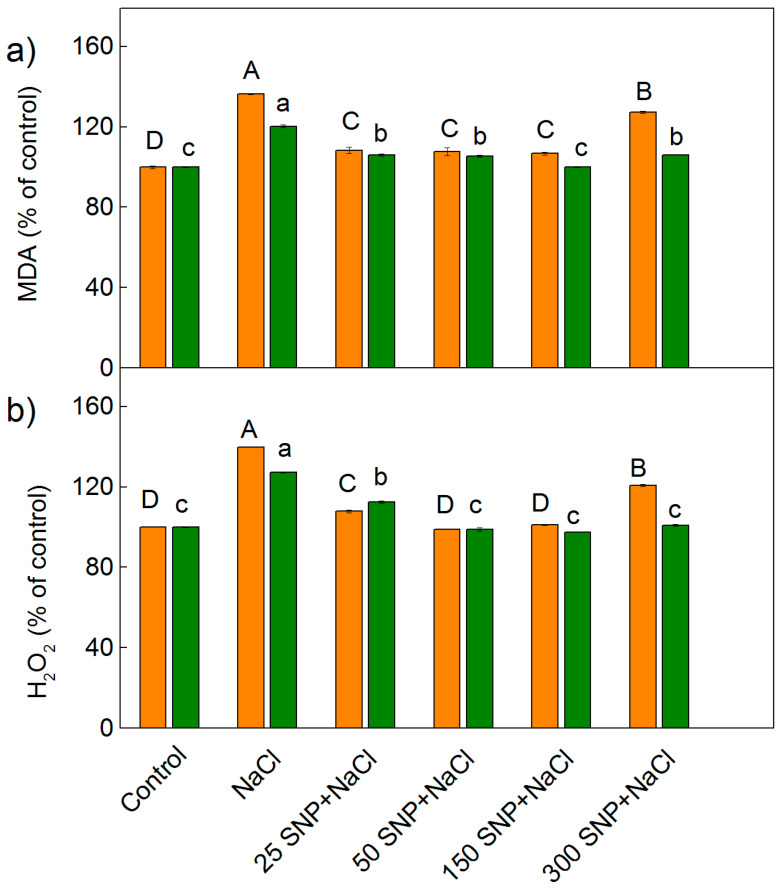
Effects of SNP on the content of MDA (**a**) and H_2_O_2_ (**b**) in the leaves of two sorghum varieties (Albanus—orange bars and Shamal—green bars) under salt stress. The values are expressed as a percentage of the control (100%). Control values for Albanus are: MDA = 231.2 nmoles g^−1^ DW, H_2_O_2_ = 132.2 nmoles g^−1^ DW; and for Shamal are: MDA = 167.0 nmoles g^−1^ DW, H_2_O_2_ = 109.6 nmoles g^−1^ DW). Mean values (±SE) were calculated from eight independent measurements. The significant differences among treatments at *p* < 0.05 are indicated by different letters (uppercase for Albanus and lowercase for Shamal).

**Figure 3 plants-12-00832-f003:**
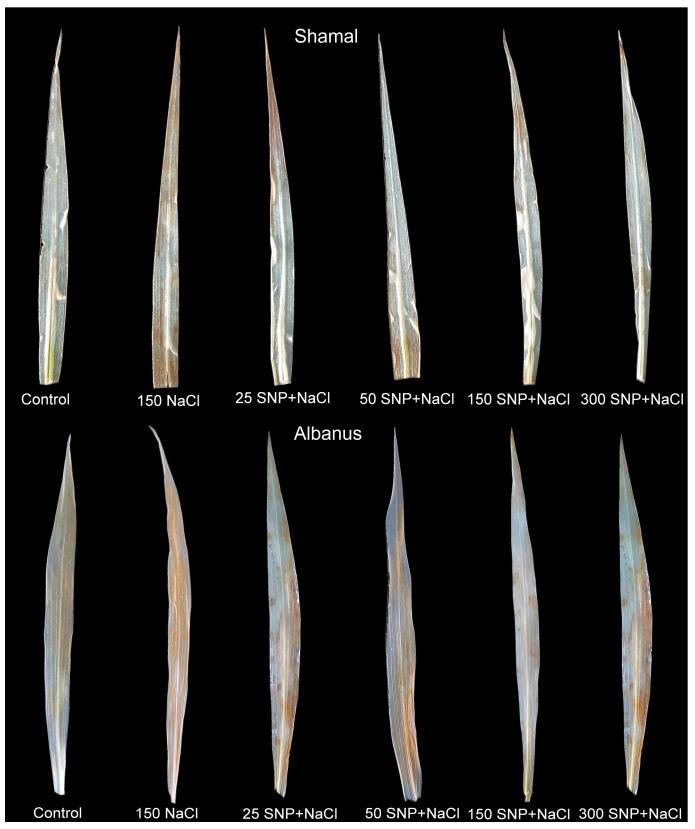
Histochemically detected accumulation of hydrogen peroxide (H_2_O_2_) with diaminobenzene (DAB) in leaves of the two sorghum varieties (Albanus and Shamal) after foliar spraying with different SNP concentrations under salt stress.

**Figure 4 plants-12-00832-f004:**
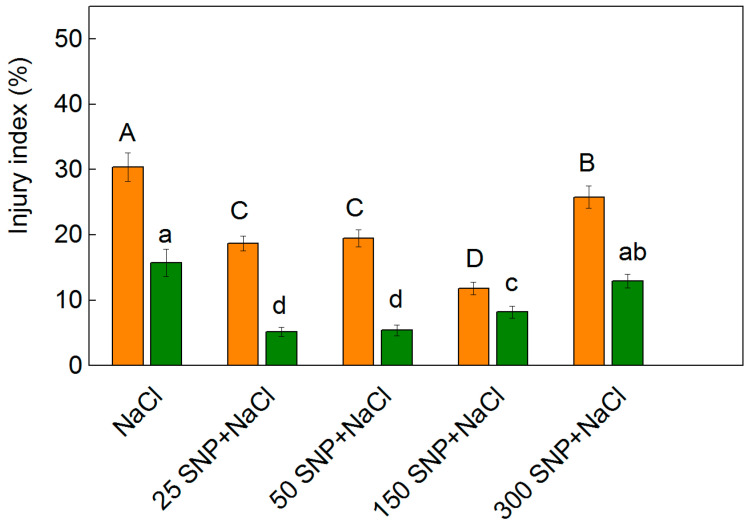
Injury index of leaves from the two sorghum varieties (Albanus—orange bars and Shamal—green bars) treated with 150 mM NaCl alone and after spraying with different concentrations of SNP (0–300 μM). Mean values (±SE) were calculated from eight independent measurements. The significant differences among treatments at *p* < 0.05 are indicated by different letters (uppercase for Albanus and lowercase for Shamal).

**Figure 5 plants-12-00832-f005:**
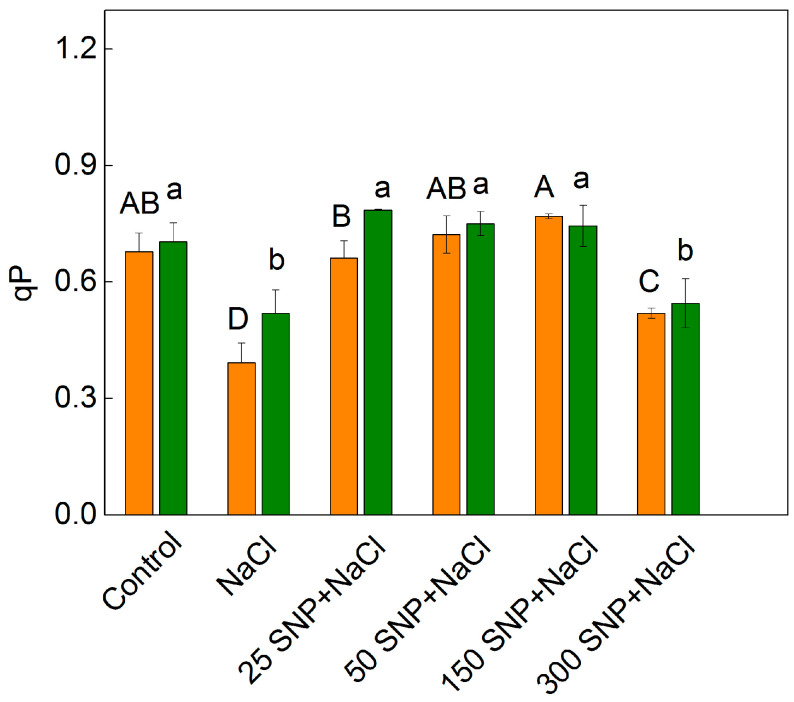
Effects of different SNP concentrations on the photochemical quenching (qP) under salt stress in the leaves of the two sorghum varieties (Albanus—orange bars and Shamal—green bars). Mean values (±SE) were calculated from eight independent measurements. The significant differences among treatments at *p* < 0.05 are indicated by different letters (uppercase for Albanus and lowercase for Shamal).

**Figure 6 plants-12-00832-f006:**
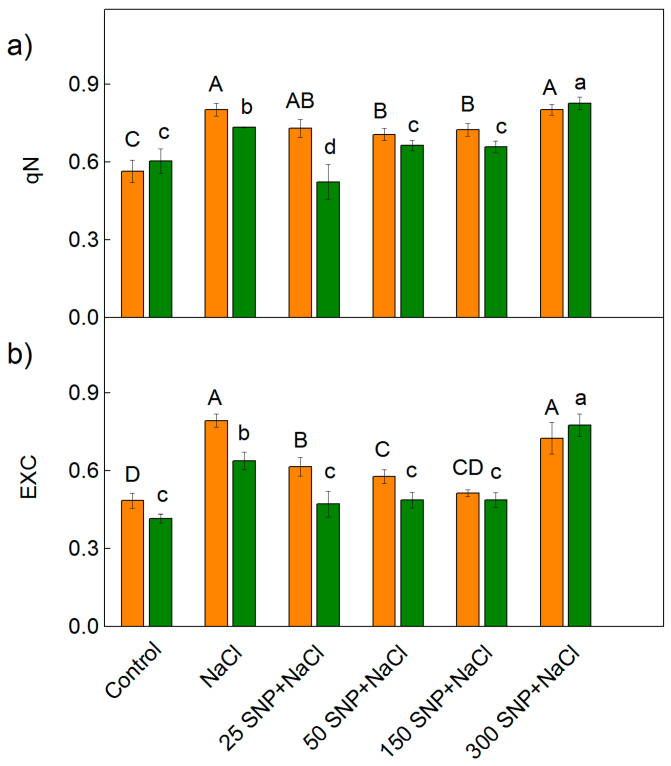
Effects of SNP on (**a**) the non-photochemical quenching of the chlorophyll fluorescence (qN) and (**b**) the excess excitation energy (EXC) in the leaves of the two sorghum varieties (Albanus—orange bars and Shamal—green bars) under salt stress. Mean values (±SE) were calculated from eight independent measurements. The significant differences among treatments at *p* < 0.05 are indicated by different letters (uppercase for Albanus and lowercase for Shamal).

**Figure 7 plants-12-00832-f007:**
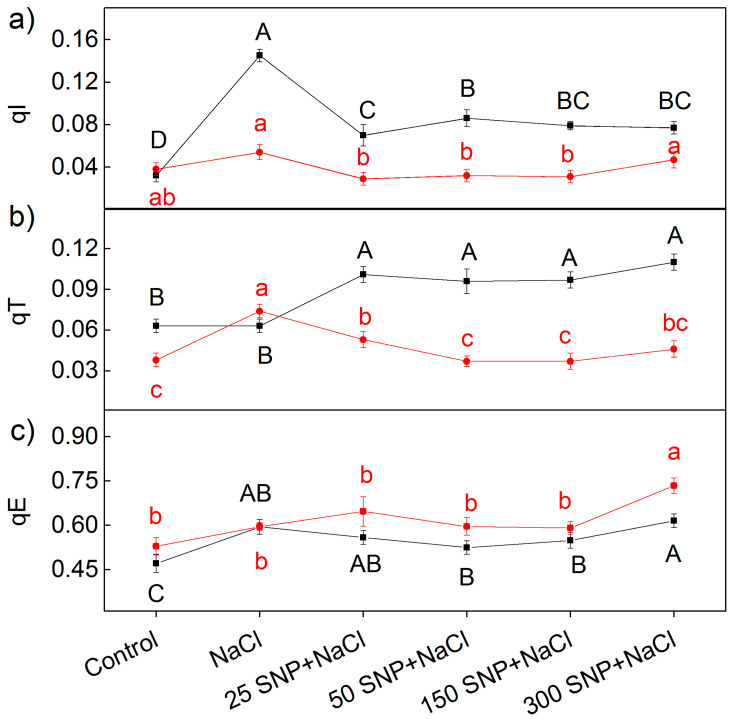
Effects of SNP on the components of non-photochemical quenching: (**a**) qI (photoinhibitory component), (**b**) qT (state-transition component), and (**c**) qE (energy-dependent component) for the leaves of the two sorghum varieties (Albanus—black and Shamal—red symbols) under salt stress. Mean values (±SE) were calculated from eight independent measurements. The significant differences among treatments at *p* < 0.05 are indicated by different letters (uppercase for Albanus and lowercase for Shamal).

**Figure 8 plants-12-00832-f008:**
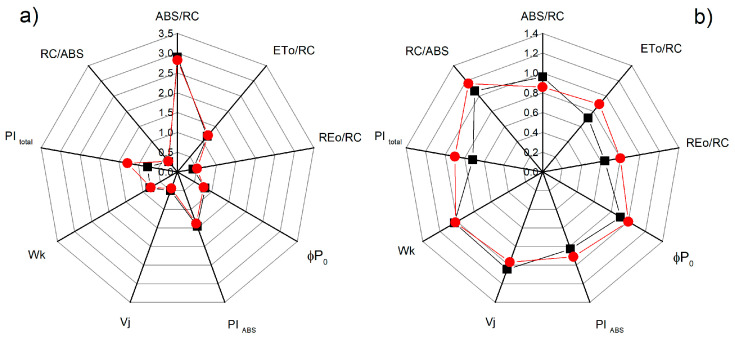
Effects of SNP on the JIP parameters in the leaves of the two sorghum varieties (Albanus—black and Shamal—red symbols) under salt stress: (**a**) at physiological conditions; (**b**) in the presence of 150 mM NaCl (data are normalized to the respective control). The following parameters are presented: the absorption flux per reaction center (ABS/RC), electron transport flux from Q_A_ to Q_B_ per PSII (ETo/RC), the electron flux reducing end acceptors at the acceptor side of PSI (REo/RC), maximum quantum yield of primary photochemistry (φPo), the performance indexes PI_ABS_ and PI_total_, relative variable fluorescence at the J step (Vj), the ratio of the K phase to the J phase (Wk), and the numbers of active reaction centers per PSII antenna chlorophyll (RC/ABS). Mean values (± SE) were calculated from 20 independent measurements.

**Figure 9 plants-12-00832-f009:**
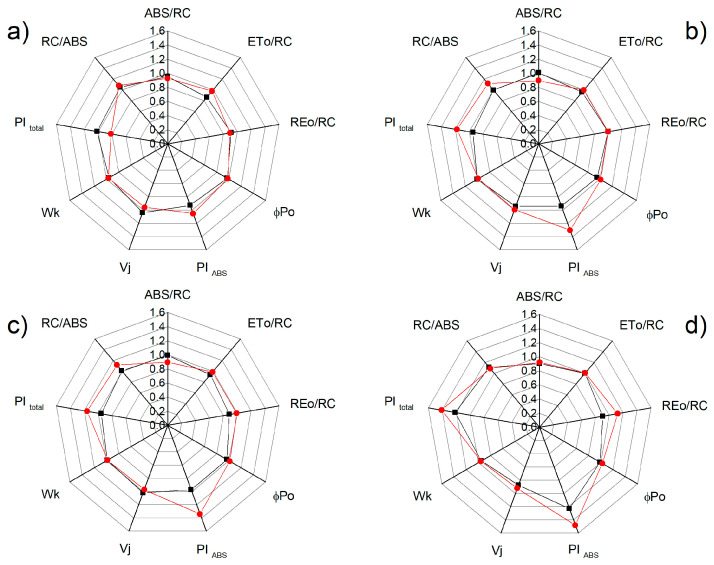
Effects of different SNP concentrations: (**a**) 25 μM SNP; (**b**) 50 μM SNP; (**c**) 150 μM SNP; (**d**) 300 μM SNP on the JIP parameters in the leaves of the two sorghum varieties (Albanus—black and Shamal—red symbols) under salt stress (150 mM NaCl). The following parameters are presented: the absorption flux per reaction center (ABS/RC), electron transport flux from Q_A_ to Q_B_ per PSII (ETo/RC), the electron flux reducing end acceptors at the acceptor side of PSI (REo/RC), maximum quantum yield of primary photochemistry (φPo), the performance indexes PI_ABS_ and PI_total_, relative variable fluorescence at the J step (Vj), the ratio of the K phase to the J phase (Wk), and the numbers of active reaction centers per PSII antenna chlorophyll (RC/ABS). Mean values (± SE) were calculated from 20 independent measurements. The parameters are normalized to the respective control.

**Figure 10 plants-12-00832-f010:**
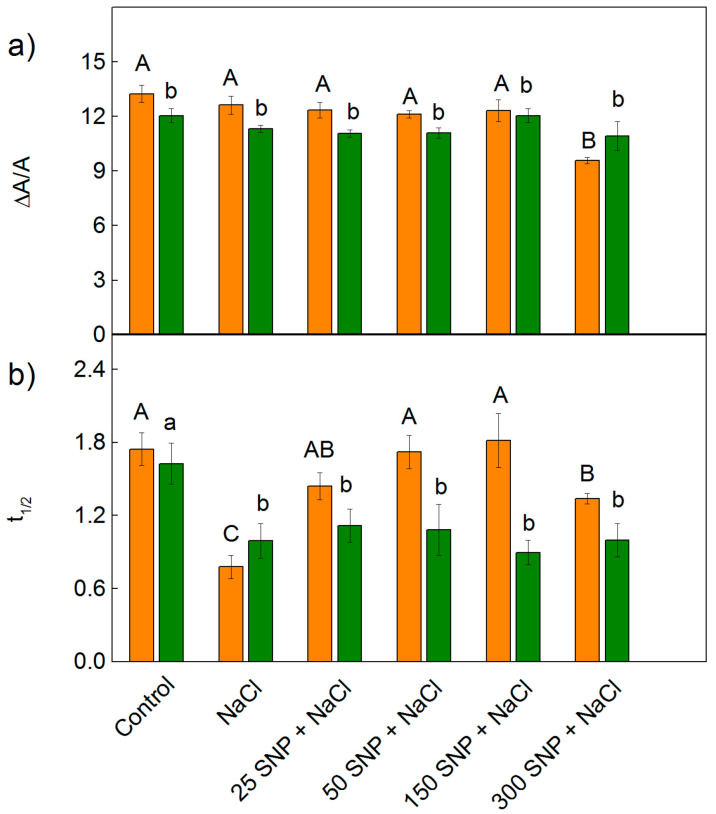
Effects of SNP on the relative changes in (**a**) the amount of P700^+^ (ΔA/A) and (**b**) half time (t_1/2_, sec) of dark reduction of P700^+^ in the leaves of the two sorghum varieties (Albanus—orange bars and Shamal—green bars) under salt stress. Mean values (±SE) were calculated from eight independent measurements. The significant differences among treatments at *p* < 0.05 are indicated by different letters (uppercase for Albanus and lowercase for Shamal).

**Table 1 plants-12-00832-t001:** NO content in leaves of two sorghum varieties (Albanus and Shamal) after treatment with different SNP concentrations and 150 mM NaCl.

Variants	NO Content (nmoles g^−1^ FW)	
	Albanus	Shamal
Control	38.49 ± 2.75 ^E^	50.18 ± 3.87 ^d^
150 mM NaCl	45.76 ± 1.33 ^D^	60.94 ± 2.66 ^cd^
25 μM SNP + NaCl	49.50 ± 1.22 ^CD^	63.32 ± 2.79 ^c^
50 μM SNP + NaCl	52.67 ± 1.35 ^BC^	71.56 ± 4.71 ^bc^
150 μM SNP + NaCl	57.72 ± 1.91 ^AB^	74.30 ± 3.83 ^b^
300 μM SNP + NaCl	63.01 ± 2.25 ^A^	94.58 ± 4.50 ^a^

The significant differences among treatments at *p* < 0.05 are indicated by different letters (uppercase for Albanus and lowercase for Shamal).

## Data Availability

Not applicable.

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
