# Peer review of "Protective Effects of Sodium Nitroprusside on Photosynthetic Performance of Sorghum bicolor L. under Salt Stress"

_plants, 2023, doi:10.3390/plants12040832_

Round 1
Reviewer 1 Report
Manuscript Review Report
According to the study of the manuscript with the number “plants-2178728” and tittle “Protective Effects of Sodium Nitroprusside on Photosynthetic Performance of Sorghum bicolor L. under salt stress” I found some short comings so the manuscript can be accepted after after doing some suggested corrections and modifications. So the authors are requested to kindly consider the following points:-
v First major issue that I faced while doing the review of this manuscript is that authors did not mention the line number, so its very difficult to correctly specify reviewer comment.
v Authors are requested to put The Materials and Methods section immediately after the Introduction section. So that readers first get an idea of the experiment and methodology used in manuscript, it will make research paper more interesting and reader friendly.
v In Introduction section 3rd paragraph , 2nd line authors had mentioned “physiological and stress condition…” it is better to rewrite the sentence as normal physiological and stress conditions.
v In Introduction section 4th paragraph, 10th line authors had mentioned “seedling age as well the duration…” Kindly rewrite it as seedling age as well as the duration…”
v In Introduction section 4th paragraph , 12th line authors had mentioned “ crop germination control ..” Kindly rewrite it as crop germination under control condition.
v In Introduction section 4th paragraph, 16th line authors had mentioned “ NO can ameliorate salt..” Kindly rewrite it as NO had ameliorated the salt ..”
v In result section 1st paragraph 5th line it is a suggestion to kindly replace the word bigger with higher, similarly at 10th line replace the word smaller with lesser.
v In result section 1st paragraph 13th line it is a suggestion to kindly replace the word prevention with any other better synonyms.
v In result section 1st paragraph 13th line authors had mentioned that “ in Shamal the prevention was registered only at 25 µM and 50 µM SNP. However, from the Figure 1b 150 µM and 300 µM concentrations are also playing an ameliorative role, although the % of amelioration may be lesser (that is the values are not reaching up to control) as compared to 25 µM and 50 µM SNP but we cannot say that only these two concentrations are showing amelioration and higher two concentrations are not, So kindly check the figure and/or data again and do the required corrections.
v In result section 2nd paragraph 7th line authors had mentioned the concentration of SNP (25µM -150 µM) my suggestion is there is no need to mention the concentration again and again, because in the next line authors have again mentioned the similar concentration of SNP, so let 7th line as general for all the tested concentration of SNP.
v In result section 4th paragraph 4th line authors are requested to recheck the sentence and rewrite it in a better way.
v In result section 4th paragraph is written two times, first before figure 3 and second time after Figure 3 so kindly delete it as per your choice.
v Figure 4 is without control, authors are requested to redraw this figure/graph and put control, whether authors are comparing the values of mentioned parameters with control or not, that is not a question (important) but It is not a correct scientific way of making graph without control. Therefore, its compulsory to put the control value in all graphs and tables.
v In result section 5th paragraph 3rd line kindly provide the abbreviation for non-photochemical quenching.
v In result section 6th paragraph 5th line replace the word “NaCl only” with “NaCl alone”.
v In result section 7th paragraph for qI parameters results, authors had only mentioned the varietal difference results but not mentioned the effect of tested concentration of SNP against control and/or NaCl. So, authors are requested to rewrite these results of qI again and try to write it in better possible way in order to describe the effect of tested concentrations also.
v As far as figures 8 and 9 are concerned I am not satisfied with the way authors draw the figures because it was very difficult to understand the results in respect to these figures.
v In result section Table 1 authors mention first values for NO content for Shamal variety and then Albanus However in rest of all the figures authors first mention Albunus then Shamal variety, my suggestion is to follow one pattern in whole manuscript to avoid any confusion for the readers and make it reader friendly and more scientific.
v Discussion section is very well written by the authors only at 4th paragraph 12th line replace word smaller with lesser or use any other synonyms of authors choice.
v In Material method section 1st paragraph 5th line authors wrote “….solution described previously [20], authors are requested to rewrite it as “….solution described previously by Stefanov el al [20].
v Similar discrepancy has been observed in 3rd paragraph 3rd line, 4th paragraph 2nd line, 6th paragraph 2nd line and 7th paragraph 9th line.
v Regarding the experimental design (1st paragraph 7th-10th line) authors had mentioned that “ After 14 days of growth, the two sorghum varieties were sprayed with different concentrations of NO donor SNP (25 μM, 50 μM, 150 μM and 300 μM) and transferred to a Hoagland nutrient solution with 150 mM NaCl..”, My question is, is stress (NaCl) and SNP foliar spray treatment was given to the plants simultaneously at the same day? It is quite strange, because, how is it possible that the plants show saline stress immediately and since the plants are receiving SNP at the same time how it will ameliorate the stress? When Both stress and SNP works at the same time then neither the toxicity would be generated properly, nor the role of SNP would be explore. What I feel is if we want to study the effect of NaCl toxicity we should provide it to the plants at least 2-3 days previous to SNP treatment, so that the plant can sense saline stress and may get the time to activate its internal defense mechanism or show stress symptoms. Kindly provide valid justification for the above-mentioned doubts.
v At most of the places In material method section I observed that the authors had mentioned the name of the scientist who did that particular parameters in his research instead of providing the root source (actual scientist who proposed that methodology) so kindly check and do the required corrections as and where required.
Author Response
Report to the comments of Reviewer 1 on manuscript (plants-2178728) titled “Protective Effects of Sodium Nitroprusside on Photosynthetic Performance of Sorghum bicolor L. under Salt Stress” by Stefanov et al.
Dear Reviewer,
The authors would like to thank you for constructive and insightful comments in relation to this work. We considered all comments and suggestions to be justified and corrected the manuscript accordingly. Please, find the detailed list of all edits below. The newly edited text parts and new references are indicated with red letters.
According to the study of the manuscript with the number “plants-2178728” and tittle “Protective Effects of Sodium Nitroprusside on Photosynthetic Performance of Sorghum bicolor L. under salt stress” I found some short comings so the manuscript can be accepted after after doing some suggested corrections and modifications. So the authors are requested to kindly consider the following points:
v First major issue that I faced while doing the review of this manuscript is that authors did not mention the line number, so its very difficult to correctly specify reviewer comment.
Answer: Sorry for the inconvenience caused.
v Authors are requested to put The Materials and Methods section immediately after the Introduction section. So that readers first get an idea of the experiment and methodology used in manuscript, it will make research paper more interesting and reader friendly.
Answer: We agree with your proposal, but according to the requirements of this Journal, Materials and Methods section should be after the Discussion and before the Conclusion.
v In Introduction section 3rd paragraph , 2nd line authors had mentioned “physiological and stress condition…” it is better to rewrite the sentence as normal physiological and stress conditions.
v In Introduction section 4th paragraph, 10th line authors had mentioned “seedling age as well the duration…” Kindly rewrite it as seedling age as well as the duration…”
v In Introduction section 4th paragraph , 12th line authors had mentioned “ crop germination control ..” Kindly rewrite it as crop germination under control condition.
v In Introduction section 4th paragraph, 16th line authors had mentioned “ NO can ameliorate salt..” Kindly rewrite it as NO had ameliorated the salt ..”
v In result section 1st paragraph 5th line it is a suggestion to kindly replace the word bigger with higher, similarly at 10th line replace the word smaller with lesser.
v In result section 1st paragraph 13th line it is a suggestion to kindly replace the word prevention with any other better synonyms.
Answer: All corrections are made in the revised manuscript.
v In result section 1st paragraph 13th line authors had mentioned that “ in Shamal the prevention was registered only at 25 µM and 50 µM SNP. However, from the Figure 1b 150 µM and 300 µM concentrations are also playing an ameliorative role, although the % of amelioration may be lesser (that is the values are not reaching up to control) as compared to 25 µM and 50 µM SNP but we cannot say that only these two concentrations are showing amelioration and higher two concentrations are not, So kindly check the figure and/or data again and do the required corrections.
Answer: Corrections are made to this paragraph in the revised manuscript.
v In result section 2nd paragraph 7th line authors had mentioned the concentration of SNP (25µM -150 µM) my suggestion is there is no need to mention the concentration again and again, because in the next line authors have again mentioned the similar concentration of SNP, so let 7th line as general for all the tested concentration of SNP.
v In result section 4th paragraph 4th line authors are requested to recheck the sentence and rewrite it in a better way.
Answer: Corrections are made in the revised manuscript.
v In result section 4th paragraph is written two times, first before figure 3 and second time after Figure 3 so kindly delete it as per your choice.
Answer: Sorry for the error that was made, this is corrected in the revised manuscript.
v Figure 4 is without control, authors are requested to redraw this figure/graph and put control, whether authors are comparing the values of mentioned parameters with control or not, that is not a question (important) but It is not a correct scientific way of making graph without control. Therefore, its compulsory to put the control value in all graphs and tables.
Answer: This is our omission, since we have not given the equation for calculating the membrane injury index (as in Kocheva et al., 2014, doi:10.1111/ppl.12074). Sorry, we have not explained the calculation of membrane injure index clearly enough. This parameter represents the relative changes after the treatment with NaCl or NaCl and SNP (T1,2) compared to control plants (C1,2) and is expressed as a percentage (i.e. the control is assumed to be zero). The equation (I (%) = [1 – (1 – T1/T2)/(1 – C1/C2)].100) is given and explained in the revised manuscript (Section 4.3.).
v In result section 5th paragraph 3rd line kindly provide the abbreviation for non-photochemical quenching.
v In result section 6th paragraph 5th line replace the word “NaCl only” with “NaCl alone”.
Answer: Corrections are made in the revised manuscript.
v In result section 7th paragraph for qI parameters results, authors had only mentioned the varietal difference results but not mentioned the effect of tested concentration of SNP against control and/or NaCl. So, authors are requested to rewrite these results of qI again and try to write it in better possible way in order to describe the effect of tested concentrations also.
Answer: Corrections in this paragraph are made in the revised manuscript.
v As far as figures 8 and 9 are concerned I am not satisfied with the way authors draw the figures because it was very difficult to understand the results in respect to these figures.
Answer: The induction curve of chlorophyll fluorescence allows the determination of many parameters, and the JIP-test parameters are usually represented in this way [ Zushi et al.,2012, . Sci. Hortic. (Amsterdam). 2012, 148, 39–46, doi:10.1016/j.scienta.2012.09.022; Esmaeilizadeh et al., 2021, PLoS One 2021, 16, e0261585, doi:10.1371/journal.pone.0261585; Dąbrowski et al., 2016, J. Photochem. Photobiol. B Biol. 157, 22–31, doi:10.1016/j.jphotobiol.2016.02.001].
v In result section Table 1 authors mention first values for NO content for Shamal variety and then Albanus However in rest of all the figures authors first mention Albunus then Shamal variety, my suggestion is to follow one pattern in whole manuscript to avoid any confusion for the readers and make it reader friendly and more scientific.
Answer: Corrections are made in the Table 1 of the revised manuscript.
v Discussion section is very well written by the authors only at 4th paragraph 12th line replace word smaller with lesser or use any other synonyms of authors choice.
v In Material method section 1st paragraph 5th line authors wrote “….solution described previously [20], authors are requested to rewrite it as “….solution described previously by Stefanov el al [20].
v Similar discrepancy has been observed in 3rd paragraph 3rd line, 4th paragraph 2nd line, 6th paragraph 2nd line and 7th paragraph 9th line.
Answer: Corrections are made in the revised manuscript.
v Regarding the experimental design (1st paragraph 7th-10th line) authors had mentioned that “ After 14 days of growth, the two sorghum varieties were sprayed with different concentrations of NO donor SNP (25 μM, 50 μM, 150 μM and 300 μM) and transferred to a Hoagland nutrient solution with 150 mM NaCl..”, My question is, is stress (NaCl) and SNP foliar spray treatment was given to the plants simultaneously at the same day? It is quite strange, because, how is it possible that the plants show saline stress immediately and since the plants are receiving SNP at the same time how it will ameliorate the stress? When Both stress and SNP works at the same time then neither the toxicity would be generated properly, nor the role of SNP would be explore. What I feel is if we want to study the effect of NaCl toxicity we should provide it to the plants at least 2-3 days previous to SNP treatment, so that the plant can sense saline stress and may get the time to activate its internal defense mechanism or show stress symptoms. Kindly provide valid justification for the above-mentioned doubts.
Answer: The plants were pretreated with SNP for 24 hours before the NaCl treatment to allow SNP to enter in the plants. Pretreatment with SNP was also applied by other authors when studying the influence of nitric oxide under salt stress [Adamu et al., 2018, Agronomy 2018, 8, 276, doi:10.3390/agronomy8120276; Yıldız et al., Eur. J. Biol. 2020, doi:10.26650/EurJBiol.2020.0026; Tahjib-Ul-Arif M. et al., 2022 (Front. Plant Sci., https://doi.org/10.3389/fpls.2022.957735 and ref. therein]. Moreover, Adamu et al., 2018 demonstrated that pretreatment with SNP has better effects on the plants under salt stress.
v At most of the places In material method section I observed that the authors had mentioned the name of the scientist who did that particular parameters in his research instead of providing the root source (actual scientist who proposed that methodology) so kindly check and do the required corrections as and where required.
Answer: Corrections are made in the revised manuscript.
Sincerely yours,
Dr. Emilia Apostolova

Reviewer 2 Report
In this study, authors studied the impacts of SNP concentrations on two sorghum varieties under salt stress, which is of practical significance and therefore the topic of the manuscript is good.
Experiments were well constructed and, therefore, the presented results are reliable.
1 Try to rewrite the abstract and conclusions, conclusions too long
2 Introduction section: Please add the relevant and latest research references
3 Figures: I have a problem with the y-coordinate units
4 Material and methods section: Please explain the composition of nutrient solution and the setting of the dark treatment time
Author Response
Report to the comments of the Reviewer 1 on manuscript (plants-2178728) titled “Protective Effects of Sodium Nitroprusside on Photosynthetic Performance of Sorghum bicolor L. under Salt Stress” by Stefanov et al.
Dear Reviewer,
The authors would like to thank you for constructive and insightful comments in relation to this work. We considered all comments and suggestions to be justified, and corrected the manuscript accordingly. Please, find the detailed list of all edits below. The newly edited text parts and new references (see reference list) are indicated with red letters.
Comments and Suggestions for Authors
In this study, authors studied the impacts of SNP concentrations on two sorghum varieties under salt stress, which is of practical significance and therefore the topic of the manuscript is good.
Experiments were well constructed and, therefore, the presented results are reliable.
1 Try to rewrite the abstract and conclusions, conclusions too long
Answer: Corrections are made in the revised manuscript.
2 Introduction section: Please add the relevant and latest research references
Answer: New relevant and latest references are added in the revision manuscript.
3 Figures: I have a problem with the y-coordinate units
Answer: Thank you for the comment. Figures 1 and 2 show the change in pigments and MDA and H2O2 compared to the corresponding untreated (control) plants. These values are expressed as a percentage of the respective control. The values of the control plants are given in the figure legends of the revised manuscript.
For the parameter presented in Figure 4, more information is given in the Materials and Methods (Section 4.3.) and Results sections of the revised manuscript. This parameter shows the changes after treatment with NaCl or NaCl and SNP compared to control plants and is expressed as a percentage.
PAM parameters are in the relative units (Figures 5 and 6) and are usually written only with the abbreviation [Zhao et al., 2019, HortScience 54(12):2125–2133, doi.org/10.21273/HORTSCI14432-19; Roháček, 2002, Photosynthetica, 40, 13–29, doi:10.1023/A:1020125719386].
4 Material and methods section: Please explain the composition of nutrient solution and the setting of the dark treatment time
Answer: Corrections are made in the revised manuscript.
Sincerely yours,
Dr. Emilia Apostolova

Round 2
Reviewer 1 Report
Dear Authors of the manuscript entitled “Protective Effects of Sodium Nitroprusside on Photosynthetic Performance of Sorghum bicolor L. under salt stress”. It was nice to see the modified version of your manuscript as you people have incorporated all the suggested recommendations. So, now the manuscript in the modified form can be accepted
